# Rehabilitation of brachial plexus injury in contact sport: Where are the data that underpin clinical management? A scoping review

Rebecca Armstrong[1], Tom McKeever[1], Michael Leavitt[1], Colin McLelland[1,2], David F. Hamilton[1,3]*

1 School of Health and Social Care, Edinburgh Napier University, Edinburgh, United Kingdom, 2 MAHD National Sports Academy, Saudi Arabia, 3 Research Centre for Health, Glasgow Caledonian University, Glasgow, United Kingdom

* david.hamilton@gcu.ac.uk

## Abstract

### Background

Although a common injury there is a lack of published primary data to inform clinical management of sports related brachial plexus injuries.

### Methods

A systematic search was completed in Medline, CINAHL, PubMed, SPORTDiscus and Web of Science databases and Google Scholar from inception to August 2023 according to the PRISMA-ScR guidelines. Methodological quality assessment of included articles was with the Joanna Briggs Institute tool. Studies providing primary data as to the rehabilitative management of diagnosed or suspected brachial plexus injuries sustained when playing contact sports were included.

### Results

Sixty-five studies were identified and screened, of which, 8 case reports were included, incorporating 10 participants with a mean age of 19.8 (±4.09) years. There was wide heterogeneity in injury severity, injury reporting, physical examination and imaging approaches documented. 9 of 10 participants returned to competitive sports, though follow-up periods also varied widely. Whilst return to play criteria varied between studies, the most consistent indicator was pain-free shoulder range of motion and strength.

### Conclusions

There is a distinct lack of data available to inform evidence-based rehabilitation management of sports related brachial plexus injury. Only 8 individual case reports contain published data reporting on 10 athletes. Further reporting is critical to inform clinical management.

**Data Availability Statement:** All relevant data are within the manuscript and its Supporting Information files.

**Funding:** The authors received no specific funding for this work.

**Competing interests:** The authors have declared that no competing interests exist.

## Introduction

Contact sports, such as American football, rugby, or wrestling, by nature, expose participants to physical trauma and can be defined as "a sport in which participants come into bodily contact with another" [1]. Tackling is an inherent part of these sports and the techniques employed typically involves contact with the opponent via the arm/shoulder, necessarily risking injury to the neck and shoulder region and, by extension, the brachial plexus.

The brachial plexus is a complex anatomical structure, comprising spinal nerves and their terminal branches in the upper extremity [2]. This includes the C5,6,7,8 and T1 spinal nerves, which provide crucial sensory and motor innervation to the muscles of the upper limb to provide normal function [3]. Brachial Plexus Injury (BPI) is comparatively rare in the general population, typically occurring in relation to road traffic accidents [4]. In these instances of high velocity impact, the patient's injuries can be significant resulting in axonotmesis (axonal damage and Wallerian degeneration) or neurotmesis (complete transection of the nerve), require surgical exploration and intervention [5, 6].

Severe BPI injuries involving axonotmesis or neurotmesis are rare in contact sports, however neuropraxia injuries (preserved axonal integrity), which are characterised by transient sensory or motor loss, are comparatively commonplace [7]. Neuropraxia related BPIs are so well known that the colloquial terminology 'stinger' and 'burner' used in common sports parlance [8].

There are three primary mechanisms of injury to the brachial plexus in contact sport; direct compression of the brachial plexus at the supraclavicular region, traction injury due to depression of the ipsilateral shoulder with concomitant side flexion of the neck to the contralateral shoulder, and cervical nerve root compression due to hyperflexion or hyperextension of the neck [9, 10]. Any mechanism can result in any severity of injury and the sequalae of these can vary hugely, depending on the degree of nerve damage, ranging from spontaneous resolution to significant functional limitation [11].

Contact sport governing bodies have made efforts to reduce the risk of cervical spine injuries through rule changes, such as in American football, where 'spear tackling' (associated with traction injury) has been banned and players now encouraged to tackle with a 'head up' position to limit neck hyperflexion [12]. Rugby administrators have also banned 'spear tackling' and are actively engaged in reducing the contact aspects of the game to mitigate serious collision-based trauma [13]. Despite this, contact sport athletes remain at risk of BPI due to the inherent tackling positions and impact forces accepted during play [10]. Multiple studies suggest a 'stinger' injury rate of around 2 per 10,000 athlete-exposures in American football [14, 15], and cohort studies have reported that more than half of American football players suffered a BPI during their career [16, 17]. Similarly, Kawasaki et al. found that in a cohort of 569 rugby players, 33% reported a history of BPI, with a re-injury rate of 37% [18]. Injury recovery periods were generally short but varied in this group, with 80% reporting full recovery the same day though 6% reported symptoms that lasted beyond 2-weeks. The wider impact of BPI is likely under-reported in this population due to the well know reluctance of athletes to self-report symptoms [14, 19].

The primary management of neuropraxia/stinger injuries is through non-operative rehabilitation. Effective rehabilitation management is paramount to minimise the risk of long-term complications or injury recurrence [20]. Accepting a range in severity of presentation, and requirements of rehabilitation based on individual case presentation, no accepted or recognised rehabilitation management protocols exist for sports related BPI [7, 21–23]. As such, the rehabilitative management of an individual BPI injury remains somewhat ambiguous, with return to play decisions often difficult for sports team medical staff to make [23]. Athletes that

report full symptom resolution are generally returned to the field at the next opportunity following this, though prolonged symptomology or recurrent injuries may trigger additional cervical imaging and wider diagnostics to inform further management [21, 24].

Local rehabilitation approaches will vary and rely on; the injury presentation, the athletes reporting of symptomology, access to clinical diagnostic imaging, and on clinician experience. The quality of the underlying evidence-base which inform the various rehabilitation interventions employed is unclear, with recent reviews of stinger management noting only generic rehabilitative techniques such as 'stretching' 'strengthening' and 'electrical stimulation techniques'[7], or nebulous concepts such as postural correction and myofascial release [9], with very limited evidence referenced to support these interventions. As such, the aim of this study was to evaluate primary data underlying rehabilitative management approaches for BPI management in contact sport.

## Methods

A scoping review of the literature was undertaken in line with the Preferred Reporting Items for Systematic Reviews and Meta-Analyses extension for scoping reviews (PRISMA-ScR) guidelines (S1 Checklist) [25]. Our protocol is available via the open science framework https://osf.io/b6ptu which included our design framework and search strategy (documented here as supplemental data).

### Information sources and search strategy

We applied the population, concept, and context (PCC) criteria to inform our search strategy that aimed to find articles that reported the rehabilitative management of brachial plexus injuries sustained while playing contact sports (S2A Table in S1 File). The search strategy was devised in conjunction with a specialist librarian and an electronic search of the following databases was conducted from inception to 21st August 2023 in Medline, PubMed, CINAHL, SPORTDiscus and Web of Science. Boolean operators were employed in the searches as detailed in the supplemental data (S2B File in S1 File). We applied an English language restriction but no other filters to the search. Manual searches of Google Scholar and citation searching of included manuscripts were also completed.

### Eligibility criteria and study selection

Due to the nature of the research question, all study types were considered for inclusion provided they were published as peer-reviewed articles and contained relevant information. We sought articles that provided primary data as to the rehabilitative management of diagnosed or suspected brachial plexus injuries sustained when playing contact sports. Contact sport was defined as a sport in which participants come into bodily contact with another [1]. All grades of competitor were considered, high school, collegiate or professional level. As we were looking for the data underpinning rehabilitative management, we excluded expert opinion or narrative articles that commented on the topic without providing underpinning data to support positions, and also material that was not available as a full text publication, such as conference abstracts.

A three-part screening strategy was employed to identify relevant articles. Two investigators independently carried out the searches and screened by title. Abstracts were reviewed independently by the same two investigators and consensus reached for full text inclusion. In the event of disagreement, or doubt, manuscripts were included for full text review. Full texts were reviewed by the same two reviewers independently and final selection agreed by consensus with a third independent reviewer.

## Data extraction and synthesis

We extracted data as to rehabilitation management and also relevant contextual data around the injury history, presentation, diagnosis, diagnostics and outcomes. The following characteristics of each study were extracted to a bespoke excel database: Author, year of publication, country of publication, study type, number of subjects, clinical presentation (mechanism of injury, injury history, physical examination), Additional imaging and diagnostic findings, diagnosis given, medical/surgical management, rehabilitative treatment interventions, return to play recommendations, follow-up timelines and outcomes. Data extraction forms were created, and 2 researchers independently extracted the data from included articles. The extracted data was cross-checked by a third researcher to ensure consistency. The case report nature of the information collected prohibited formal pooling of data. As such the results are presented descriptively.

## Quality assessment

To assess internal validity and risk of bias, the Joanna Briggs Institute (JBI) critical appraisal tool for case reports was utilised [26]. The tool comprises of an 8-point checklist addressing study design and reporting, with: Yes, No, Unclear and Not applicable selection options for each component. There are no accepted thresholds for case report study inclusion within a systematic review, [27] however, Dekkers et al. [28] emphasise that the completeness of this quality assessment tool relates to case report reliability. We report the results in this context.

# Results

The literature search generated one-hundred and thirty-three articles (Medline:27; PubMed: 15; CINAHL:19; SPORTDiscus:32; Web of Science:35; Google Scholar:5) Following the removal of duplicates, 65 papers were evaluated against the eligibility criteria. After screening, only nine were eligible for full text review, with the majority of exclusions being expert opinion articles. One article was not accessible due to a broken hyperlink, and a further two were conference abstracts. Seven of these were eligible for inclusion alongside a single further article that was found through citation searching of the included publications, bringing the final number of included publications to eight. Full details are displayed in the PRISMA flowchart (Fig 1).

## Study characteristics

All the included articles were case reports. One article discussed three individual cases [29], bringing the total number of individuals included in the eight articles to ten athletes. The mean age of subjects was 19.8 years (±4.09). Five individual participants were American football players. [11, 29–32] two were wrestlers, [29, 33] two were rugby union players, [34, 35] and one was a basketball player [29]. Six of the eight reports were from America, [11, 29–33] one from New Zealand [35] and one from Italy [34].

## Mechanism of injury and symptomology

Injury characteristics and presentation timelines varied (Table 1). The most commonly stated injury mechanisms were traction (n = 4) [11, 31–33] and compression(n = 3) [29, 30, 35] the three other reports detailing no clear mechanism. Five participants were reported to have sustained recurrent BPI injuries, [11, 30–32, 34] while five were first time presentations [29, 33, 35]. The symptomology recorded primarily incorporated burning pain and altered sensation in the upper limb, [11, 29–31, 33, 34] alongside motor weakness in the upper limb [11, 29–31,

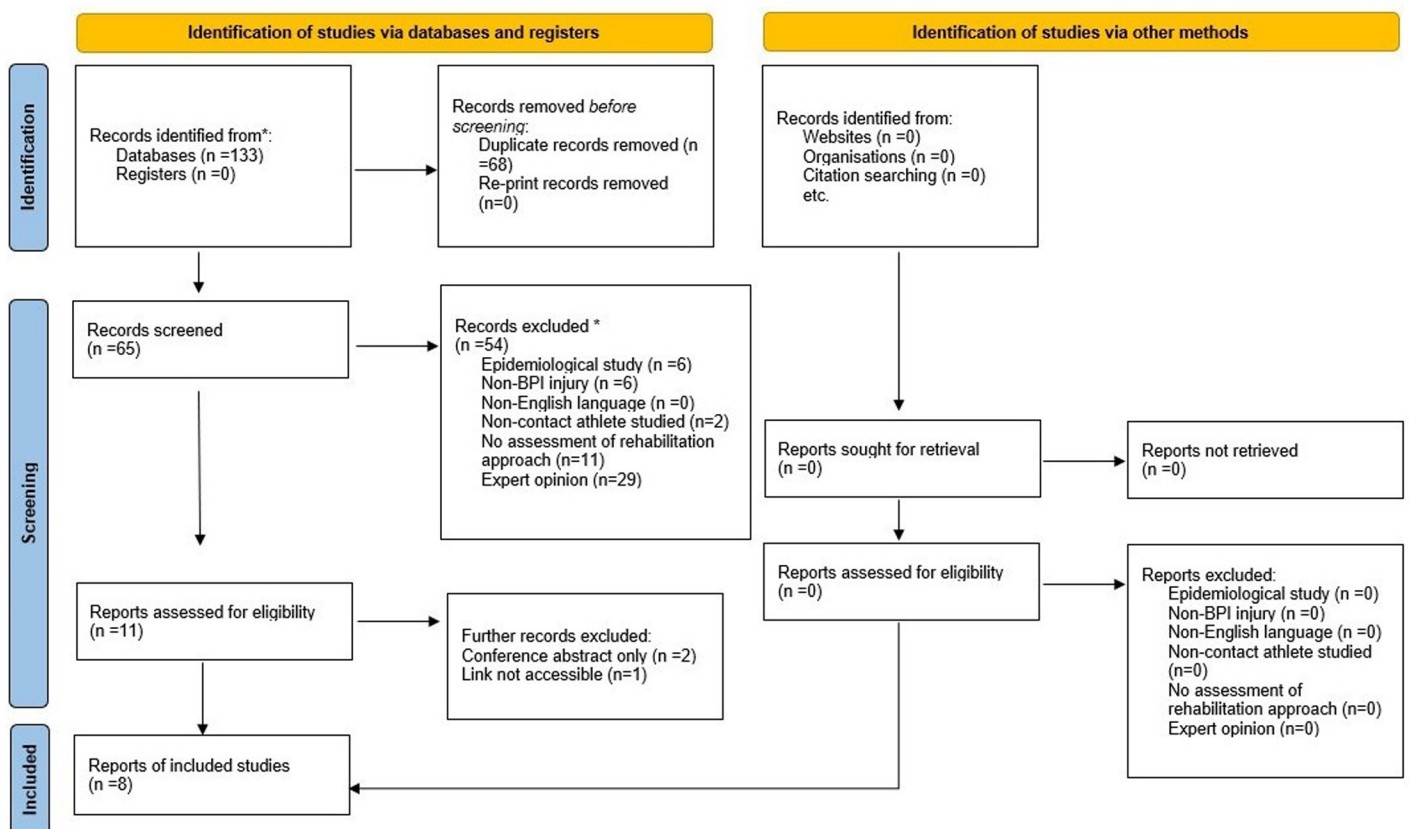

**Fig 1. PRISMA flow chart.**

33–35]. Traction injuries caused biceps brachii motor weakness in all 4 cases [11, 31, 33, 34] and muscle atrophy in the deltoid region was reported in 3 of 4 cases, [11, 29, 34] whereas compression injuries led to rotator cuff weakness in all 3 cases [29, 30, 35].

## Imaging, diagnoses and surgical interventions

Imaging modalities were reported in seven of the ten cases (Table 1). Radiographs in 4/10, [30, 31, 33, 35] MRI in 7/10, [11,30–35] CT myelogram in 1/10 [11] and arthrogram in 1/10 [29]. EMG reports were generated for 6/10 [11, 29, 32, 34] participants and abnormal nerve conduction in the upper limb musculature was noted in all of these cases. Saliba et al. [11] utilised the EMG reports to guide rationale for surgical intervention, while the remaining studies utilised EMG reports to guide RTP and for injury prognostics [29, 32, 34].

The injuries and diagnoses reported differed between all case, and were described variously as; 'Recurrent stinger injuries', [32] 'Brachial plexus neuropraxia', [30] 'Avulsion of C5 and C6 nerve roots', [11] 'Grade 2 Burner', [31] 'Acute Brachial Plexus Neuropathy', [29] 'Postfixed brachial plexus', [33] 'Brachial plexus injury', [35] Traumatic paresis of the axillary nerve [34]. Two cases required surgical intervention [11, 33]. with the rest receiving primary rehabilitative management [29–32, 34, 35].

**Table 1. Injury history, examinations and diagnosis.**

| Author & Year | Article type | Participants | Injury history and physical examination | Diagnostic imaging | Diagnostic terminology | Surgical intervention |
|---|---|---|---|---|---|---|
| Zaremski et al., (2017), USA | Case Report n = 1 | 16-year-old male American Football player | **Mechanism:** Not stated<br>**Injury history:** 13x previous BPI episodes reported.<br>**Physical Examination:** No examination stated for presenting episode. Previous examination showed Cervical Active and Passive ROM normal, neck strength normal in all planes and normal neurological examination | **Imaging:** No imaging for presenting injury. Previous results; Torg ratios normal, relative spinal stenosis = canal diameter of 10 mm at the C3/4 level<br>**EMG:** Mild chronic bilateral neurogenic changes at superior trunk of the brachial plexus | Recurrent Stinger Injuries | No |
| Hartley and Kordecki, (2018), USA | Case Report n = 1 | 17-year-old American football player | **Mechanism:** Tackling in head down position, compression to superior aspect of shoulder. Returned to game where acute symptoms returned in another tackle attempt.<br>**Injury history:** 6x previous BPI episodes reported.<br>**Physical Examination:** Dull, burning pain in left lateral aspect of neck and left shoulder. Altered sensation in left bicep brachii, exacerbated by active and passive right cervical side-bending. Noted sharp pain in posterior cervical and left periscapular regions with active cervical extension, weakness in scapular and cervical stabilizers and C6-7 myotome. Tightness throughout the cervical spine and presumed tightness of pectoralis minor. | **Imaging:** Cervical MRI and plain radiographs showed loss of cervical lordosis, intervertebral discs normal height and alignment<br>**EMG:** None | Brachial plexus Neuropraxia | No |
| Saliba et al., (2009), USA | Case Report n = 1 | 19-year-old American Football player | **Mechanism:** Attempted tackle, opposing players helmet striking players upper chest and shoulder causing contralateral cervical side flexion.<br>**Injury history:** 5x previous BPI episodes reported.<br>**Physical Examination:** Numbness of the left arm. No Cervical pain or tenderness. No motor function elicited from the left shoulder or bicep, but motor function rapidly developed in hand and wrist. Within 30 minutes severe burning pain in left C5-C6 dermatome. Left upper trapezius tender on palpation. AROM absent in the left shoulder and elbow flexion, AROM at the hand and wrist improved quickly. Grip strength remained in the left hand, with reduced power through finger extension, abduction, thumb extension, wrist flexion and extension. Reduced triceps, biceps, anterior deltoid, pectoralis major power. No power elicited through the middle and posterior deltoid and rotator cuff musculature. In days following injury, atrophy of the deltoid, Shoulder subluxed with visual sulcus sign. | **Imaging:** extensive brachial plexus injury on chest MRI (Cervical normal). Absent C5 and C6 nerve roots on CT myelogram.<br>**EMG:** 3-weeks post Injury, abnormal sensory responses at or proximal to the dorsal root ganglion in both C5 and C6. No evidence of C5-6 upper trunk innervation, with normal function of the rhomboids A complete preganglionic lesion at C6. 4 months post-surgery, developing potentials in the bicep and deltoid, but no activity in the suprascapular nerve. | Avulsion of C5 and C6 nerve roots | Multiple nerve root transfers |

*(Continued)*

**Table 1.** (Continued)

| Author & Year | Article type | Participants | Injury history and physical examination | Diagnostic imaging | Diagnostic terminology | Surgical intervention |
|---|---|---|---|---|---|---|
| Nissen et al., (1996), USA | Case Report n = 1 | 15-year-old American Football player | **Mechanism:** 2x sequential neck injuries playing American football. Injury 1- forceful contralateral cervical flexion to right hand side- left arm felt paralyzed and numb, symptoms resolved within 5-minutes, did not seek medical attention. Injury 2- 'head on contact' at 7-days following injury 1, caused sever shooting pains down left neck and arm and persistent shoulder weakness. **Injury history:** 2x previous BPI episodes reported. **Physical Examination:** 1-week post 2nd injury, inability to accept resistance left supraspinatus with mild weakness of left deltoid and biceps and inability to accept full resistance, and diminished left biceps and brachioradialis deep tendon reflexes. Positive drop arm test, negative Spurling test and Hawkins sign, and equivocal Neer impingement test. Active range of movement of 80° of abduction, 70° of external rotation, 160° of flexion and full internal rotation. Sensory testing with pin prick and light touch unremarkable. | **Imaging:** No bony injury on cervical radiographs with normal MRI and CT studies (performed due to persistent neurologic symptoms). **EMG:** None | 'Grade 2 Burner' | No |
| Kuzma et al., (2013), USA | Case Report n = 1 | 23-year-old male wrestler | **Mechanism:** Traction injury to left upper extremity during wrestling match. Immediate pain in neck and shooting pain in left upper extremity with lasting numbness and weakness of the posterior aspect. Did not seek medical attention, but completed match and 1x additional match with no change in symptoms. **Injury history:** discectomy of L5-S1 disc herniation secondary to axial-load injury 2 years prior to presenting condition **Physical Examination:** 6-days post injury as symptoms had not resolved. Numbness and tingling in posterior left upper extremity continuing distally to the fingers, with weakness in shoulder and elbow movement. Cervical active and passive movement pain limited, Full movement of shoulder and elbow. Muscle weakness left biceps and triceps. Positive brachial plexus traction test though negative Spurling's test and cervical compression test = negative. 16-days post injury, symptoms continued to persist with Spurling's test now positive. | **Imaging:** No bony abnormalities on radiographs. Left paracentral disc herniation at T1–T2 with mild to moderate central narrowing and moderate left lateral recess narrowing of the vertebral canal. **EMG:** None | Posterolateral herniation of the T1–T2 disc impinging on the T1 nerve root, with the post fixed brachial plexus resulting in symptoms of C7 radiculopathy. | Surgical excision of the T1–T2 intervertebral disc and T1–T2 lamino-foraminotomy |

(*Continued*)

**Table 1.** (Continued)

| Author & Year | Article type | Participants | Injury history and physical examination | Diagnostic imaging | Diagnostic terminology | Surgical intervention |
|---|---|---|---|---|---|---|
| Reid & Trent (2002), New Zealand | Case Report n = 1 | 25-year-old male professional rugby player | **Mechanism:** Left side blunt supraclavicular trauma during tackle, experienced electric shock type pain radiating down radial border of left arm to thumb and index finger. Symptoms resolved within 30-60s, did not leave the field of play.<br>**Injury history:** None reported.<br>**Physical Examination:** Post game, complained of persistent paraesthesia of left thumb and index finger. Mild left sided weakness in shoulder external rotation, elbow flexion, wrist extension. Mild left-sided supraclavicular tenderness. Full shoulder movement with negative apprehension test. Biceps brachialis reflex reduced. Days following injury, the player developed discomfort of extension and left sided lateral flexion with tenderness on palpation of left of C6 vertebrae. | **Imaging:** No bony abnormalities on radiograph. No disc related radiculopathy or regional cervical stenosis on MRI<br>**EMG:** None | Brachial plexus injury involving the left sided (C6) nerve root | No |
| Frizziero et al., (2018), Italy | Case report n = 1 | 27-year-old male professional rugby player | **Mechanism:** Traction injury. Reported to be unable to throw a ball.<br>**Injury history:** Previous auxiliary nerve traumatic injury and lesion 1-year prior.<br>**Physical examination:** Burning symptoms and paraesthesia into deltoid region. Full shoulder movement noted, however pain reported with overhead movements. Deltoid weakness. Reduced deltoid tone and mass of whole deltoid region. External rotation and abduction resisted muscle power tests were reported to be positive. Positive O'Brien test. Mild glenohumeral joint instability noted with Grade 3 scapular dyskinesia. | **Imaging:** Increased MRI signal intensity at C5 and denervation oedema noted at infraspinatus.<br>**EMG:** traumatic paresis of the axillary nerve and denervation of the deltoid muscle. | Traumatic paresis of axillary nerve. Sunderland grade III-IV and chronic C5 myotome sufferance. | No |
| Hershman et al., (1989), USA | Case Series n = 3 | 23-year-old basketball player | **Mechanism:** No clear mechanism stated. Right shoulder soreness following playing basketball. Awoke following morning with constant burning pain in right deltoid, P + subsided after 2/52, noted limitation in shoulder movements<br>**Injury history:** None reported<br>**Physical Examination**: 2-months post onset, atrophy of right shoulder gridle, profound weakness of right deltoid and supraspinatus. Moderate winging of Right scapula. Decreased sensation to light touch along the axillary nerve. | **Imaging:** Arthrogram negative<br>**EMG:** low amplitude right axillary motor nerve and right lateral antebrachial cutaneous nerve response. Severe motor unit potential loss in right deltoid, supraspinatus, infraspinatus, and serratus anterior | Acute Brachial Plexus Neuropathy | No |
| | | 17-year-old male wrestler | **Mechanism:** Onset of left shoulder pain during wrestling, no specific trauma noted<br>**Injury history**: None reported<br>**Physical Examination:** 4 -weeks post, 'slight' atrophy of the left deltoid, biceps, triceps and wrist extensors. | **Imaging:** None<br>**EMG:** Normal upper limb sensory nerve and motor nerve conduction studies. Modest numbers of fibrillation potentials in the left deltoid, infraspinatus, supraspinatus, biceps, pronator teres, and brachioradialis | Not Stated | No |

(*Continued*)

**Table 1.** (*Continued*)

| Author & Year | Article type | Participants | Injury history and physical examination | Diagnostic imaging | Diagnostic terminology | Surgical intervention |
|---|---|---|---|---|---|---|
| | | 16-year-old male American football player | **Mechanism:** 'Struck' on right shoulder by another player's helmet during practice<br>**Injury history:** None reported<br>**Physical Examination:** 1-day post-onset, weakness with pain on right shoulder abduction, internal rotation and external rotation, sensation normal. Resolved by 2-weeks | **Imaging:** None<br>**EMG:** 1 month after onset, normal sensory and motor nerve conduction studies with minimal fibrillation potentials and motor unit potential loss in the infraspinatus and nonspecific motor unit potential changes in the supraspinatus and deltoid | Not stated | No |

## Treatment and rehabilitation approach

In all cases, treatment was by both acute and rehabilitative management phases, however the reporting of acute timeframes differed between cases (Table 2). Acute interventions varied, but entailed soft tissue inflammation management, including: cold therapies, [11, 31, 33] rest, [29, 32, 33] use of a hemi-sling, [11, 30] soft tissue therapies [31, 33, 35] and cervical mobilisation [30]. All four cases reporting traction injuries applied cold therapy and rest and/or contraindication of strengthening exercises [11, 33, 34]. All reports describing the management of compression injuries however reported the use of strengthening as an acute treatment approach [29, 30, 35]. Only one study reporting a compression injury reported the use of self-stretching as an acute treatment approach [30]. The management approaches described in the subsequent rehabilitation stage also varied, encompassing: maintenance of cardiovascular fitness, [31, 33, 35] strengthening of shoulder musculature [29–31, 33–35] strengthening of cervical musculature, [31, 32] and scapular stabilisation exercises [31, 34].

## Follow-up, outcomes and return to play recommendations

There was no consistent approach across the included papers towards follow-up timelines, outcome reporting or return to play guidance (RTP) (Table 2). The follow-up timescales reflected assessments from 16-days to 5-years post-injury. 1 study did not state a follow-up report [30]. Objective improvements in clinical presentation were generally noted across the case studies, and nine of ten individuals returned to unrestricted participation in sport [29–35]. RTP recommendations and the criteria for RTP clearance varied between studies. Pain-free shoulder range of motion and strength was the most commonly used indicator for the RTP decision [29–31].

## Quality assessment

The methodological quality of the studies varied (Table 3). 4/8 studies met all 9 of the evaluation criteria, [11, 31, 33, 34] and a further 3/8 met at least 75% of items [29, 30, 35]. 1 study failed to reach 50% of the reporting criteria [32]. Due to the limited availability of published primary research data, the decision was made to include all of these case studies in the review irrespective of reporting quality. Out with the case-report reporting quality assessment evaluation, the complexity of the brachial plexus injury characteristics and variation in management approaches resulted in poorly generalisable data.

## Discussion

Despite the potential for severe and debilitating outcomes resulting from BPI in the sporting athlete, there is a distinct lack of evidence supporting rehabilitation management approaches

Table 2. Rehabilitation management, return to play and outcome.

| Author & Year | Treatment Approach | Return to play Recommendation | Follow Up | Outcome |
|---|---|---|---|---|
| Zaremski et al. 2017 | Acute treatment of pain control, rest, and restriction from competition. Active rehabilitation only when asymptomatic at rest with focus on cervical ROM, posture, and muscular imbalances. If symptoms continue/ evidence of foraminal narrowing and/or cervical disc disease on MRI, planned use of fluoroscopically guided epidural steroid injections for pain relief or even surgical decompression of a narrowed foramen or fusion may be indicated for continued weakness. | Approved by Orthopaedic Surgeon with recommendation of the use of Kerr Collar to reduce acceleration and force transmission through the neck. | Not Stated | Completed the next two seasons without further injury. |
| Hartley et al. 2018 | Specific progressive rehabilitation described: Week 1 (x3/week), self-stretching, muscle activation, mobilisations, periscapular strengthening, deep anterior cervical strengthening, avoid contralateral side bending in early stages. Week 2 (3x/week), continue home exercises, stretching and periscapular/ deep cervical strengthening, cervical extension exercises. Week 3–5 (2x week) continue HEP stretching and strengthening, overhead and forward press strengthening in cervical extension, tackling replication in 'heads up position'. | Approved by physician to return to full contact sports including football upon demonstration of full symptom-free strength and ROM of the cervical spine and upper extremity, with normal Cervical plain-film radiographs. However also recommended discontinue football if any further episodes of BPN occurred. | After 10 rehab visits (5-weeks), regained full pain-free active cervical extension, symptom-free contralateral side-bending, normal scapular strength, and 'proper head and neck positioning' maintained during tackling replication. | Not Stated |
| Saliba et al. 2009 | Acute treatment (daily) of ice to supraclavicular area, passive movements of elbow and shoulder, functional tasks for hand and wrist. Hemi-sling used to prevent subluxation of shoulder. Post-op rehabilitation commenced after swelling and pain subsided. 3x weekly passive movement and active-assisted movement of the shoulder using pulleys. Stretch reflex to facilitate motor function of bicep. Movement and manual resistance of wrist, hand and triceps. Desensitization techniques in paresthetic and dysesthetic regions, light neural mobilisations. Interrupted direct current used to stimulate deltoid, biceps, brachialis, supraspinatus, infraspinatus, and pectoralis major until time of reinnervation. | N/A | 4-months, pain free and improvements in elbow flexion to 50˚, shoulder abduction 10˚, hypesthesia remained in thumb. 12 months, strength improvements noted. Able to perform bicep curls with 7 lbs of weight, bench press 30 lbs. Shoulder abduction against gravity 60˚ | Preclude any sport participation due to moderate disability of left upper extremity |
| Nissen et al. 1996 | Acute treatment of ice massage of upper trapezius and shoulder, PRICE principle. Strength training considered contraindicated in acute phase if neurogenic muscle weakness present, stated that EMG data should be collected prior to strength training commencing. Progressive rehabilitation to maintain cardiovascular fitness, active-resisted ROM exercises for cervical spine and shoulder. Isometric exercise for Cervical strengthening. Following demonstration of full shoulder movement, isometric and closed-kinetic-chain strengthening initiated, progressing to isotonic and isodynamic strengthening of shoulder external and internal rotators, abductors, extensors and scapular stabilisers, and elbow flexion and extension. | Physician advised not to return to football for the rest of the season. Planned resumption of competition when full strength and ROM of the upper extremity and neck returns, and EMG findings are negative for acute de-innervation and no increased insertional activity, fibrillation potentials, or positive sharp waves. | 1-month, Minimal weakness on MMT of the external rotators and deltoid. Deep tendon reflexes were symmetrical. Occasional paraspinal neck pain persisted, full pain free Cervical ROM, Spurling test was negative, and arm pain absent. 2-months, strength and deep tendon reflexes were normal. Pain resolution. | Able to play competitive basketball 4-months after initial injury. |
| Kuzma et al. 2013 | Nonoperative management with ice, hot packs, massage, electrical stimulation, shortwave diathermy, and over-the-counter nonsteroidal anti-inflammatory drugs (noted minimal symptom improvement). Post-operative management: Avoidance of all activity for 10-days until suture removal and wound had healed. Subsequent rehabilitation: cardiovascular activity, rotator cuff strengthening, triceps extensions, and lateral raises using resistance bands. Rapid early progression added push-ups, chin-ups, dips, seated and bent rows, and dumbbell chest presses with increasing repetitions and weight daily. | Based on clinical examination, the neurosurgeon released the patient to participate fully without restrictions in all wrestling activities. | 18-days post-surgery, neurosurgical follow up. Noted negative Spurling's test, no pain, parathesis, or reproducible symptoms and normal strength. | 31-days post-op, participated in competition 'without difficulty', winning weight class. The individual competed at the national tournament 'without symptoms' 6-weeks post-op. |

*(Continued)*

**Table 2.** (Continued)

| Author & Year | Treatment Approach | Return to play Recommendation | Follow Up | Outcome |
|---|---|---|---|---|
| Reid et al. 2002 | Acute treatment (1–10 Days) of NSAID and acupuncture to relive pain and soft tissue therapy and mobilisations to restore cervical spine mobility. Further rehabilitation, continued to run and cycle, and participate in 'non-contact' drills plus elasticated band strengthening program for rotator cuff. | Not stated | 5-weeks reported full recovery. 'Minimal improvements' noted over first 3-weeks. | Returned to participating in full contact Rugby at 5-week. |
| Frizziero et al. 2018 | Rehabilitation consisted of 4 land-based exercise sessions and 2 water-based exercise sessions per week for 2-months. Acute phase (1–30 days): Strengthening exercises for serratus anterior/ trapezius/ rhomboids, Stabilisation of the glenohumeral joint through rotator cuff strengthening exercises. Recovery exercises for abduction and external rotation of the shoulder and whole deltoid strengthening. Second phase (30–60 days): Return to play focused rehabilitation. Restoration of motor and postural control. Velocity conditioning, proprioception, and velocity/ explosive strength training. Conservative rehabilitation plan with introduction of electrostimulation testing–Reporting to be H 2 h-b FES sessions of 45 minutes a day based in balanced triangular biphasic waves (200-10ms, 20 Hz, 0-30mA). | As able and by symptoms. Return to training with gradual introduction of tackling and contact play. | 2-months, Pain resolution. Deltoid muscle reported to be normotrophic/ normal scapular kinesis reported with no evidence of upper trapezius compensatory movement. Negative rotator cuff tests and shoulder stability intact. EMG tests showed full innervation of deltoid with reduced voluntary recruitment. | Returned to full play with no recurring symptoms. |
| Hershman et al. 1989 | Generalised treatment reported across participants for acute brachial neuropathy. Acute phase (initial period from onset of symptoms until resolution of pain); extremity is rested, analgesics for pain control, sling in severe cases to protect extremity. Rehabilitation phase: Strengthening of denervated muscles. Trunk-scapula relationship considered and serratus anterior and rhomboid involvement. | Considered when athlete reaches plateau in strength and recovery. Suggests that strength parity may be difficult to achieve, thus must be considered on a case-by-case basis | 4-years, weakness persisted in deltoid and external rotators. EMG: suggestive of chronic neurogenic change in supraspinatus, infraspinatus, and serratus anterior Lateral deltoid electronically silent- long standing denervation without reinnervation of the muscle | Able to play basketball, did note some occasional shoulder pain |
| | | | 5-years, 'occasional sense of weakness' in shoulder on heavy lifting. Resolution of pain, paraesthesia, and anaesthesia. 'Mild' weakness in anterior deltoid. EMG normal. | Unrestricted in wrestling participation |
| | | | 2-years, 'occasional fatigue' in shoulder after heavy lifting. Examination showed 'traces of scapula winging' and 'mild' weakness in posterior deltoid. EMG normal. | Did not return to football for the remainder of the season but returned following year. |

or the rationale for setting return to play criteria. Through systematic review of the published primary data for BPI rehabilitation in contact sport, only eight reports, representing ten individual case studies were found. No trials, cohort studies, or even retrospective registry-based studies are available to inform clinical management, which then, necessarily, is driven by expert opinion and the application of basic rehabilitation principles.

**Table 3. JBI quality assessment [26].**

| | Frizziero etal., (2018) | Hartley and Kordecki, (2018) | Hershman etal., (1989) | Kuzma etal., (2013) | Nissen etal., (1996) | Reid and Trent, (2002) | Saliba etal., 2009 | Zaremski etal., (2017) |
|---|---|---|---|---|---|---|---|---|
| Were patient's demographic characteristics clearly described? | Y | Y | Y | Y | Y | Y | Y | Y |
| Was the patient's history clearly described and presented as a timeline? | Y | Y | Y | Y | Y | Y | Y | Y |
| Was the current clinical condition of the patient on presentation clearly described? | Y | Y | Y | Y | Y | Y | Y | N |
| Were diagnostic tests or assessment methods and the results clearly described? | Y | Y | Y | Y | Y | Y | Y | N |
| Was the intervention(s) or treatment procedure(s) clearly described? | Y | Y | Y | Y | Y | U | Y | U |
| Was the post-intervention clinical condition clearly described? | Y | Y | U | Y | Y | Y | Y | N |
| Were adverse events (harms) or unanticipated events identified and described? | NA | N | NA | NA | NA | NA | Y | N |
| Does the case report provide takeaway lessons? | Y | Y | Y | Y | Y | Y | Y | Y |

There is a substantial difference in scale as to the management approaches required for differently presenting brachial plexus injuries. Transient 'stingers' may be isolated events that essentially self-resolve with no need for active treatment, whereas severe neural injuries can require significant medical intervention. Conservative management and rehabilitation will be the typical intervention for mild-modest neuropraxia based injury, whereas the most severe cases involving neural compromise or separation may require surgical intervention and result in disability. The 10 case reports we found reflected this range of presentation.

No two studies applied the same diagnostic terminology making it hard to draw parallels across the cases. Standardisation of injury evaluation and documentation would facilitate pooling of future data to inform clinical management. Variation was also seen in clinical assessment approaches, the use of imaging/ EMG, and in the diagnostic terminology applied. Interestingly, the use of EMG was reported in a number of cases, and used to determine nerve innervation recovery and to inform return to play decisions. It is unlikely that the use of nerve conduction evaluation is reflective of 'routine' return to play management following BPI for contact sports players outside of elite sports settings, or those being treated through specialist centres. This may be the result of 'interesting cases' being reported in the literature, that may not reflect the more routine situation in clinical practice. This lack of context, and potentially quite limited generalisability, results that caution must applied as to the representativeness of the pooled findings we present for the scant literature in this area.

Despite heterogeneity in injury and in the utilised assessment diagnostics, an interesting finding of this review is differential presentation and subsequent rehabilitative management of compression and traction-based injury. Compression injuries resulted in rotator cuff weakness, whilst traction injuries were associated with biceps brachii weakness. In the acute phases of treatment, traction injuries were managed conservatively, including cold therapy, rest and/ or the contraindication of strengthening exercises, whilst athletes with compression injuries were encouraged to participate in active rehabilitation at an earlier stage. Rehabilitative management follows clinical presentation and this likely reflects clinicians treating what they find, as opposed to any specific rationale for differential management of different injury patterns, as this is not otherwise reported.

We are unable to comment on the injury mechanisms leading to BPI, as details as to impact received and the setting of this are scantly reported. There are no specific notes of adverse events or reactions to rehabilitation management. Reported outcomes were generally positive and athletes returned to play following rehabilitation with no ongoing issues in 9 of 10 cases recorded. The exception being the single case involving nerve root evulsion at C5/C6, which, despite surgery, resulted in disability that impeded return to sports and substantially affected the individual's quality of life. Follow-up timeframes of the cases varied substantially though, and was often limited to a few months following injury. A narrative 'return to play' was reported, with little objective context around this statement. Rather ambiguous terminology was also used in relation to the RTP criteria applied, an example being the restoration of 'normal' cervical strength [4, 29].

Whilst wider literature suggests a comparatively high prevalence of BPI in the contact sport athlete, primary data as to how to manage this injury remains unpublished or unavailable to the scientific community and practising therapists. As highlighted by multiple authors, underreporting is a concern, and whilst transient symptoms may be a factor, one must question the rigour of current sporting injury data capture and reporting. BPIs will be managed at pitchside and then in sports and community rehabilitation settings, or they will require medical escalation. From the lack of published data found, it seems no one is publishing the data and progress of these athletes following intervention. Injury reporting remains unpopular for high profile players and teams, who often rely on specific individuals to play when injured or not at their physical peak. Whilst injury data is closely guarded by sporting teams, for fear of competitive disadvantage, the lack of any published data restricts developments and improvements in sporting injury management. Specialist sports treatment centres or national sporting programmes are also likely to collect the BPI injury data that we sought in our review and the scientific community would benefit from publication of injury management and outcomes. The current situation is that rehabilitation professionals lack evidence-based diagnostic criteria, intervention guidelines, reporting guidelines, clinical outcome selection, and return to play criteria for the range of BPI presentations in collision sports.

## Limitations

There are various limitations to this work. Despite wide review of four major databases and Google Scholar using pragmatic search terms and a rigorous review methodology, it is possible that relevant articles were missed. We suggest that any further such primary data as to the rehabilitation of sports related BPI is well-hidden, and not readily available to the practicing clinician. Case studies were the only sources that contained relevant data as to rehabilitation interventions for BPI. While disappointing, this is a major finding of this review. As case study data is relied on, we accept the risk that various selection and reporting biases may have influenced the cases presented and rehabilitation themes discussed. The ten case reports included in this review regarded the management of contact sports athletes, with seemingly enhanced access to diagnostic testing, assessment and treatment, it is likely these do not well represent of the wider experience of amateur athletes. Further, these may not be truly representative of the wider management of BPI in contact sport, as published case reports usually pertain to particularly interesting or challenging presentations, or those employing particular diagnostic or treatment technologies. Notably, eight of the ten cases included were reports from the United States of America and health care management may not be widely generalisable.

## Conclusion

Although thought to be a relatively common injury, there is a lack of consensus guidance as to the clinical management of sports related brachial plexus injuries. The published primary data

as to rehabilitation of sports related BPI is very poor, essentially consisting of 8 case reports relating to 10 individuals. Further data reporting is critical to inform clinical management. Alongside vastly more data, standardised methods of assessment, diagnostic testing, outcome evaluation and reporting across the spectrum of severity of BPI presentation are all needed to facilitate rehabilitation and return to play guidance.

## Supporting information

**S1 Checklist. Preferred Reporting Items for Systematic reviews and Meta-Analyses extension for Scoping Reviews (PRISMA-ScR) checklist.**
(DOCX)

**S1 File. Search strategy.**
(DOCX)

## Acknowledgments

We express our gratitude to Maria King, specialist librarian, Edinburgh Napier University, for her assistance with the search parameters.

## Author Contributions

**Conceptualization:** Rebecca Armstrong, David F. Hamilton.

**Data curation:** Tom McKeever, Michael Leavitt.

**Formal analysis:** Rebecca Armstrong.

**Investigation:** Rebecca Armstrong.

**Methodology:** Rebecca Armstrong, Tom McKeever, Colin McLelland, David F. Hamilton.

**Project administration:** Rebecca Armstrong, David F. Hamilton.

**Supervision:** Colin McLelland, David F. Hamilton.

**Visualization:** Tom McKeever, Michael Leavitt.

**Writing – original draft:** Rebecca Armstrong.

**Writing – review & editing:** Tom McKeever, Michael Leavitt, Colin McLelland, David F. Hamilton.

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
