## [Decision Letter · Decision Letter 0]

4 Oct 2023

PONE-D-23-27297Rehabilitation of brachial plexus injury in contact sport: Where are the data that underpin clinical management? A scoping reviewPLOS ONE

Dear Dr. Hamilton,

Thank you for submitting your manuscript to PLOS ONE. After careful consideration, we feel that it has merit but does not fully meet PLOS ONE’s publication criteria as it currently stands. Therefore, we invite you to submit a revised version of the manuscript that addresses the points raised during the review process.

We look forward to receiving your revised manuscript.

Kind regards,

Esedullah Akaras

Academic Editor

PLOS ONE

Journal Requirements:

Reviewers' comments:

Reviewer's Responses to Questions

**Comments to the Author**

1. Is the manuscript technically sound, and do the data support the conclusions?

Reviewer #1: Yes

Reviewer #2: Partly

2. Has the statistical analysis been performed appropriately and rigorously? 

Reviewer #1: Yes

Reviewer #2: I Don't Know

3. Have the authors made all data underlying the findings in their manuscript fully available?

Reviewer #1: Yes

Reviewer #2: Yes

4. Is the manuscript presented in an intelligible fashion and written in standard English?

Reviewer #1: Yes

Reviewer #2: No

5. Review Comments to the Author

Reviewer #1: The care of brachial plexus injuries (BPI) in athletes, in my opinion, is a seriously under-discussed topic in sports medicine. By primarily relying on a small number of case studies for information, it successfully highlights the lack of comprehensive data and generally accepted best practices. This demonstrates the urgent requirement for more thorough research projects and established methods for diagnosis, assessment, and outcome reporting in BPI rehabilitation for sports-related injuries. however there are following issues about the manuscript:

Abstract: The background section is notably concise and fails to provide substantial information about the context and significance of the study.

Method: It's rather intriguing that, out of the numerous articles retrieved, only case reports met your inclusion criteria. Case reports generally constitute a lower level of evidence. Could you elaborate on the criteria used for selecting these reports while excluding others? Additional clarification on this aspect would be beneficial.

Consider adding a reference to define "contact sport."

Result & Discussion:

While the section mentions variations in injury mechanisms and presentations, it does not delve into a detailed discussion of these variations or potential factors contributing to them.

The section mentions adverse events but does not provide comprehensive details or analysis of these events, which could be valuable for understanding potential risks associated with certain treatments.

Reviewer #2: Dear authors

I have thoroughly reviewed your article on addressing deficiencies in the rehabilitation and clinical management of brachial plexus injuries in contact sports. I appreciate your efforts in this specific field. It is well-known that the incidence and severity of BPI varies based on the sport played. Depending on the degree of the injury's damage, the rehabilitation procedure varies. Many athletes return to sport without requiring postgame treatment.

Although the title of your study seems to be the rehabilitation of BPI, when I read the abstract and the entire article, I saw that your primary goal is to draw conclusions about the entire rehabilitation process and determine the criteria for return to sports. Despite the tight relationship between these two issues, they are distinct and must be addressed separately. If determining the clinical management and rehabilitation process is the primary objective, the chosen keywords are insufficiently specific. Because, on reviewing the entire article, I noticed that you included physiotherapy applications among the treatment methods. The study protocol that includes keywords also incorporates physiotherapy practices as keywords. Additionally, numerous other terms were used during the search. This indicates that the decision regarding keyword selection was inadequate. If the keywords are determined and chosen appropriately, you will be able to access more studies such as . S Reid et al.‘’ Brachial plexus injuries—report of two cases presenting to a sports medicine practice’’.

In the article's introduction, injuries are categorized as severe or moderate. What do the terms severe and moderate mean? What is the frequency of these types of injuries? Even if it is in separate sports branches, it is necessary to mention it in order to inform the reader. Expressions such as uncommon and frequent provide insufficient information to the reader.

In paragraphs 79 and 80, is the phrase "In sports settings, BPIs are common enough for the slang terms'stinger' and 'burner' to be used?" a unique thought of the authors?

Writing the introduction section in a more comprehensible, fluent and holistic manner will keep the reader's interest in the article alive.

In the imaging, diagnosis, and surgical interventions section, the RTP abbreviation was used without its meaning being specified. Similarly, table abbreviations should also be defined.

Considering all of these circumstances

1- New and accurate keywords must be selected.

2- The purpose of the investigation must be determined thoroughly.

3- The introduction section should be written more clearly, fluently, and comprehensively.

4- The results and discussion sections should be rewritten based on the analysis of newly accessible articles.

I look forward to reviewing your article once these significant corrections have been made.

6. PLOS authors have the option to publish the peer review history of their article (what does this mean?). If published, this will include your full peer review and any attached files.

Reviewer #1: **Yes: **Sahar Boozari

Reviewer #2: No

---

## [Author Response · Author response to Decision Letter 0]

16 Dec 2023

thank you, please note that we have returned a fulsome response to the reviewers as a separate file. 

we would like to highlight the poor quality technical review from reviewer2 that we have had to rebut at length.

---

## [Decision Letter · Decision Letter 1]

4 Jan 2024

PONE-D-23-27297R1Rehabilitation of brachial plexus injury in contact sport: Where are the data that underpin clinical management? A scoping reviewPLOS ONE

Dear Dr. Hamilton,

Thank you for submitting your manuscript to PLOS ONE. After careful consideration, we feel that it has merit but does not fully meet PLOS ONE’s publication criteria as it currently stands. Therefore, we invite you to submit a revised version of the manuscript that addresses the points raised during the review process.

We look forward to receiving your revised manuscript.

Kind regards,

Esedullah Akaras

Academic Editor

PLOS ONE

Journal Requirements:

Reviewers' comments:

Reviewer's Responses to Questions

**Comments to the Author**

1. If the authors have adequately addressed your comments raised in a previous round of review and you feel that this manuscript is now acceptable for publication, you may indicate that here to bypass the “Comments to the Author” section, enter your conflict of interest statement in the “Confidential to Editor” section, and submit your "Accept" recommendation.

Reviewer #1: All comments have been addressed

Reviewer #2: All comments have been addressed

2. Is the manuscript technically sound, and do the data support the conclusions?

Reviewer #1: Yes

Reviewer #2: Yes

3. Has the statistical analysis been performed appropriately and rigorously? 

Reviewer #1: Yes

Reviewer #2: Yes

4. Have the authors made all data underlying the findings in their manuscript fully available?

Reviewer #1: No

Reviewer #2: Yes

5. Is the manuscript presented in an intelligible fashion and written in standard English?

Reviewer #1: Yes

Reviewer #2: Yes

6. Review Comments to the Author

Reviewer #1: The term "ambiguous" in relation to rehabilitative management (lines 120-122) needs a brief explanation for clarity. Further details or examples of local rehabilitation protocols mentioned in lines 118-122 would enhance the reader's understanding.

The rationale behind excluding expert opinion articles should be briefly explained for transparency.

Reviewer #2: (No Response)

7. PLOS authors have the option to publish the peer review history of their article (what does this mean?). If published, this will include your full peer review and any attached files.

Reviewer #1: **Yes: **Sahar Boozari

Reviewer #2: No

---

## [Author Response · Author response to Decision Letter 1]

15 Jan 2024

Review Comments to the Author

Reviewer #1: 

The term "ambiguous" in relation to rehabilitative management (lines 120-122) needs a brief explanation for clarity. Further details or examples of local rehabilitation protocols mentioned in lines 118-122 would enhance the reader's understanding. The rationale behind excluding expert opinion articles should be briefly explained for transparency.

Reviewer #2: (No Response)

Authors comments:

Many thanks for highlighting these 3 points, we have addressed these and updated the enclosed manuscript version.

Many thanks for your constructive feedback to our paper.

Author team

---

## [Decision Letter · Decision Letter 2]

23 Jan 2024

Rehabilitation of brachial plexus injury in contact sport: Where are the data that underpin clinical management? A scoping review

PONE-D-23-27297R2

Dear Dr. Hamilton,

We’re pleased to inform you that your manuscript has been judged scientifically suitable for publication and will be formally accepted for publication once it meets all outstanding technical requirements.

Kind regards,

Esedullah Akaras

Academic Editor

PLOS ONE

Additional Editor Comments (optional):

Reviewers' comments:

Reviewer's Responses to Questions

**Comments to the Author**

1. If the authors have adequately addressed your comments raised in a previous round of review and you feel that this manuscript is now acceptable for publication, you may indicate that here to bypass the “Comments to the Author” section, enter your conflict of interest statement in the “Confidential to Editor” section, and submit your "Accept" recommendation.

Reviewer #1: All comments have been addressed

2. Is the manuscript technically sound, and do the data support the conclusions?

Reviewer #1: Yes

3. Has the statistical analysis been performed appropriately and rigorously? 

Reviewer #1: Yes

4. Have the authors made all data underlying the findings in their manuscript fully available?

Reviewer #1: Yes

5. Is the manuscript presented in an intelligible fashion and written in standard English?

Reviewer #1: Yes

6. Review Comments to the Author

Reviewer #1: (No Response)

7. PLOS authors have the option to publish the peer review history of their article (what does this mean?). If published, this will include your full peer review and any attached files.

Reviewer #1: **Yes: **Sahar Boozari

---

## [Editor Report · Acceptance letter]

29 May 2024

PONE-D-23-27297R2 

PLOS ONE

Dear Dr. Hamilton, 

I'm pleased to inform you that your manuscript has been deemed suitable for publication in PLOS ONE. Congratulations! Your manuscript is now being handed over to our production team.

Kind regards, 

on behalf of

Dr. Esedullah Akaras 

Academic Editor

PLOS ONE